# Non-Linear Pharmacokinetics of Oral Roscovitine (Seliciclib) in Cystic Fibrosis Patients Chronically Infected with *Pseudomonas aeruginosa*: A Study on Population Pharmacokinetics with Monte Carlo Simulations

**DOI:** 10.3390/pharmaceutics12111087

**Published:** 2020-11-12

**Authors:** Cyril Leven, Sacha Schutz, Marie-Pierre Audrezet, Emmanuel Nowak, Laurent Meijer, Tristan Montier

**Affiliations:** 1EA 3878, GETBO, Université de Bretagne Occidentale, F-29200 Brest, France; 2CHRU de Brest, Service de Biochimie et de Pharmaco-Toxicologie, Hôpital de la Cavale Blanche, F-29200 Brest, France; 3Univ Brest, INSERM, EFS, UMR 1078, GGB, F-29200 Brest, France; sacha.schutz@chu-brest.fr (S.S.); marie-pierre.audrezet@chu-brest.fr (M.-P.A.); tristan.montier@univ-brest.fr (T.M.); 4CHRU de Brest, Service de Génétique Médicale et Biologie de la Reproduction, Centre de Référence des Maladies Rares “Maladies Neuromusculaires”, Hôpital Morvan, F-29200 Brest, France; 5Centre d’Investigation Clinique, INSERM CIC-1412, Hôpital de la Cavale Blanche, CHRU Brest, 29609 Brest CEDEX, France; emmanuel.nowak@chu-brest.fr; 6ManRos Therapeutics, Presqu’île de Perharidy, 29680 Roscoff, France; meijer@perha-pharma.com

**Keywords:** cystic fibrosis, roscovitine, Seliciclib, pharmacokinetics, Monte Carlo simulation

## Abstract

Roscovitine (Seliciclib), a new protein kinase inhibitor, was administered orally to adult patients with cystic fibrosis for the first time in the ROSCO-CF trial, a dose-escalation, phase IIa, randomized, controlled trial. Extensive pharmacokinetic sampling was performed up to 12 h after the first oral dose. Roscovitine and its main metabolite M3 were quantified by liquid chromatography coupled with tandem mass spectrometry. The pharmacokinetics analyses were performed by non-linear mixed effects modelling. Monte Carlo simulations were performed to assess the impact of dose on the pharmacokinetics of oral roscovitine. Twenty-three patients received oral doses ranging from 200 to 800 mg of roscovitine and 138 data points were available for both roscovitine and M3 concentrations. The pharmacokinetics was best described by a two-compartment parent-metabolite model, with a complex saturable absorption process modelled as the sum of Gaussian inverse density functions. The Monte Carlo simulations showed a dose-dependent and saturable first-pass effect leading to pre-systemic formation of M3. The treatment with proton-pump inhibitors reduced the rate of absorption of oral roscovitine. The pharmacokinetics of oral roscovitine in adult patients with cystic fibrosis was non-linear and showed significant inter-individual variability. A repeat-dose study will be required to assess the inter-occasional variability of its pharmacokinetics.

## 1. Introduction

Cystic fibrosis (CF) is a rare genetic disease caused by mutations in the CF transmembrane conductance regulator (CFTR) chloride channel gene [1]. Improved nutritional status, respiratory physiotherapy, and the development of modulating drugs capable of increasing the amount and number of CFTRs have drastically increased the survival of CF patients [2].

CFTR mutations are classified into six functional categories [3], which gives hope that pharmaceutical therapies specific to particular mutants can be developed. The first success, ivacaftor (VX770), was approved by the Food and Drug Administration (FDA) in 2012 for treatment of the 4 to 5% of patients who present the Gly551Asp mutation in CFTR and later for patients with other mutations in which the protein reaches the plasma membrane but does not open appropriately.

In patients with the Gly551Asp mutation, ivacaftor corrects the sweat chloride defect, improves pulmonary function and patient-reported respiratory symptoms, and results in substantial weight gain [4]. Nowadays, several ivacaftor-based formulations have been developed. The last one, a triple therapy named TRIKAFTA^®^ combining ivacaftor, tezacaftor (VX661), and elexacaftor (VX445), has proven to be superior to previous treatments [5], being also effective in CF patients heterozygous for F508del.

As the disease progresses, however, colonization and infection of the lungs by bacteria that are multi-resistant to antibiotics, in particular *Pseudomonas aeruginosa*, limit the effectiveness of antibiotic treatments for lung infections and threaten patient survival [6]. New treatments capable of slowing down this process are needed. (R)-roscovitine (roscovitine, or seliciclib) is a protein kinase inhibitor currently being evaluated as a potential therapeutic agent for CF patients (reviewed in [7]). Pre-clinical data indicate that roscovitine interferes with the proteolytic degradation of F508del-CFTR [8], and stimulates bactericidal activity in alveolar macrophages by restoring intra-phagolysosome acidic pH [9].

Roscovitine was administered to adult CF patients for the first time in the ROSCO-CF trial, a dose escalation, phase IIa, multicenter, randomized, controlled trial of CF patients with two CF-causing mutations, including at least one F508del-CFTR mutation, chronically infected with *Pseudomonas aeruginosa* (NCT02649751).

We present here the results of population pharmacokinetic (PK) analyses and Monte Carlo simulations of roscovitine and its carboxylate metabolite, M3, conducted to characterize the complex absorption process of roscovitine and to evaluate the inter-individual variability of its pharmacokinetics.

## 2. Materials and Methods

### 2.1. Investigational Drug

Roscovitine (ManRos Therapeutics, Roscoff, France) is a 2,6,9-substituted purine analogue, and it is a white to off-white crystalline solid. With an octanol/water partition coefficient of 3.11, roscovitine is freely soluble in ethanol, practically insoluble in water at pH 7, and slightly soluble in 0.01 M hydrochloric acid. In the ROSCO-CF clinical study, roscovitine was administered as an anhydrous base in hard gelatin capsules.

### 2.2. Study Design and Population

ROSCO-CF was a multicenter, randomized, controlled, phase IIA, dose-ranging trial of roscovitine in CF patients chronically infected with *Pseudomonas aeruginosa* (NCT02649751). Patients were recruited in 12 CF clinical trial centers in France and met the following inclusion criteria: age over 18 on the date of informed consent, diagnostic of CF, carrier of 2 CF causing mutations with at least one F508del-CFTR mutation, forced expiratory volume at 1 s ≥ 40%, and chronic lung *Pseudomonas aeruginosa* infection. Roscovitine was administered orally as 200 mg capsules in a single dose in the morning for four cycles of four consecutive days, separated by a 3-day treatment-free period, for a total of 4 weeks.

### 2.3. Sample Collection and Analysis

On the first day of treatment, 2 mL blood samples were drawn on heparin lithium collection tubes before dosing and at 0.5, 1, 2, 4, 6, 8, and 12 h post-dose. All blood samples were centrifuged, divided in two aliquots in polypropylene tubes and frozen for storage at −80 °C pending sample analysis. Plasma samples were analyzed for their concentration in roscovitine and its carboxylate metabolite M3 by means of a validated liquid chromatography coupled with tandem mass spectrometry (LC-MS/MS) (SCIEX, Framingham, MA, USA) method. Details of the assay method are available as Appendix A. The lower limit of quantification (LLOQ) was 5.00 ng/mL for roscovitine and its metabolite M3. To avoid bias due to different molecular weights of roscovitine and its metabolite M3, 354.45 g·mol^−1^ and 368.43 g·mol^−1^, respectively, measured concentrations were converted to nmol·L^−1^ and administered doses were converted to µmol.

### 2.4. Genotyping

CYP2B6, CYP3A4, and ABCB1 were genotyped by next generation sequencing (NGS) using the Ion System PGM™ (Thermo Fisher Scientific, Waltham, MA, USA). Library preparation was performed with the AmpliSeq Library Kit 2.0 Ion and the AmpliSeq™ Ion Pharmacogenomics Research Panel (Thermo Fisher Scientific, Waltham, MA, USA). Data analysis was run using the Torrent software Suite™ (Thermo Fisher Scientific, Waltham, MA, USA) and the final report was generated by the PGxAnalysis plugin.

### 2.5. Pharmacokinetic Model Development and Evaluation

Roscovitine and M3 PK data were analyzed using a nonlinear mixed-effect model as follows:

Y_ij_ = f(θ_ij_, t_ij_) + g(θ_ij_, t_ij_) ε_ij_(1)
where Y_ij_ is the concentration of either roscovitine or its carboxylate M3 measured for patient i at time j, the error model is ε_ij_ ~ N(0,1), θ_ij_ = h(μ, η_i_, β, z_i_), µ is the population values of the parameters, z_i_ the covariates for the patient i, β the coefficient associated with these covariates, and η_i_ the individual random effect.

The stochastic approximation expectation-maximization (SAEM) algorithm [10] implemented in the nonlinear mixed-effect modelling software MonolixSuite2019R1 (Lixoft, Orsay, France) was used for the estimation of the parameters. Plasma concentrations below the LLOQ were left censored according to the M3 method described by Beal [11]. The model building steps were managed by Sycomore (Lixoft, Orsay, France).

The structural models tested were one- and two-compartment parent-metabolite models, with first-order transformation of roscovitine in its carboxylate M3, without back transformation, and first-order elimination. To describe the complex absorption of roscovitine and its metabolite M3 satisfactorily different input models were tested for both roscovitine and M3. The first models investigated were first-order absorption (Equation (2)) and first-order absorption with lag time (Equation (3)):(2)$$f\left(t\right)={\mathrm{dosek}}_{a}e^{-k_{a}t}$$
(3)$$\{\begin{array}{c} 0,t<t_{\mathrm{lag}}\\ {\mathrm{dosek}}_{a}e^{-k_{a}\left(t-t_{\mathrm{lag}}\right)},t⩾t_{\mathrm{lag}} \end{array}$$
where f(t) is the input rate into the central compartment as a function of time (t), k_a_ is the absorption rate constant, and t_lag_ is the absorption lag-time.

To reproduce multiphasic and late-absorption processes, the additional models tested were the transit compartment model [12], Michaelis–Menten kinetics [13], the gamma distribution function, and a sum of inverse Gaussian distribution function (Equation (4)) [14,15] without subsequent first-order absorption.
(4)$$\mathrm{IR}\left(t\right)=\pi_{1}\times {\mathrm{IG}}_{1}+\pi_{2}\times {\mathrm{IG}}_{2}$$
where IR(t) is the input rate function, IG_1_ and IG_2_ the inverse gaussian distributions and π_1_ and π_2_ are the weights associated with each input fraction. The weight π_2_ was set as (1 − π_1_). Density IG_g_(t) was modelled by the following equations):(5)$${\mathrm{IG}}_{g}\left(t\right)=\sqrt{\frac{{\mathrm{MAT}}_{g}}{2{\pi \mathrm{CV}}_{g}^{2}t^{3}}}\times \exp\left(\frac{{\left(t-{\mathrm{MAT}}_{g}\right)}^{2}}{2{\mathrm{CV}}_{g}^{2}{\mathrm{MAT}}_{g}t}\right)$$
with
(6)$${\mathrm{MAT}}_{g}=\frac{T_{g}^{\max}}{\sqrt{1+\frac{9}{4}{\mathrm{CV}}_{g}^{4}}-\frac{3}{2}{\mathrm{CV}}_{g}^{2}}$$
where MAT_g_ corresponds to the mean absorption time of the g-th absorption process. The parameters T_g_^max^, and CV_g_ correspond, respectively, to the time that the g-th inverse Gaussian function reaches its maximum and its coefficient of variation.

To explore the influence of the administered dose on the proportion of roscovitine undergoing a first-pass effect, the observed oral bioavailability of roscovitine was defined according to the equation:(7)$$F_{\mathrm{dose}}=\frac{\mathrm{Oraldose}}{\mathrm{Oraldose}+D_{50}}$$
where D_50_ is the oral dose of roscovitine for which F_dose_ is 50%. Conversely, the proportion of roscovitine undergoing a first-pass effect leading to the influx of carboxylate M3 in the central compartment was defined as F_M3_ = (1 − F_dose_).

The inter-individual variability was described by an exponential model:(8)$$\theta_{i}=\theta_{\mathrm{pop}}\times \exp\left(\eta_{i}\right)$$
where θ_pop_ is the typical value of the parameter in the population and η_i_ the individual effect. The distribution of the parameters was log-normal, and logit-normal with bounds 0 and 1 for the weights associated with each input fraction. Several models of the error function g(θ_ij_, t_ij_) from Equation (1) were tested—an additive model where g=a, a proportional model where g=b×f, and a combined model such as $g=\sqrt{a^{2}+b^{2}\times f^{2}}$. The influence of different covariates was tested to determine if they could explain some of the inter-individual variability. The covariates considered for inclusion in the model were the age in years, the sex, the weight in kg, the height in cm, the glomerular filtration rate (GFR) estimated by the modification of diet in the renal disease (MDRD) formula in mL·min⁻^1^, the concomitant use of proton pump inhibitors (PPI) or of known CYP3A4 inhibitors, and the polymorphism of CYP2B6, CYP3A4, or ABCB1. The influence of covariates was modelled as follows:(9)$$\ln\left(\theta_{i}\right)=\ln\left(\theta_{\mathrm{pop}}\right)+\beta_{\mathrm{CAT}}$$
for categorical covariates. Where θ_pop_ is the typical value of the parameter in the population, θ_i_ the value of the individual parameter influenced by the Boolean covariate CAT, and β the coefficient associated to the covariate. The continuous covariates were modelled as follows:(10)$$\ln\left(\theta_{i}\right)=\ln\left(\theta_{\mathrm{pop}}\right)+\beta \times \ln\left(\frac{\mathrm{COV}}{{\mathrm{COV}}_{\mathrm{ref}}}\right)$$
where COV_ref_ is a reference value of the covariate in the population (e.g., a reference weight of 70 kg). The covariates were tested if they were graphically correlated with the fixed effects. They were retained in the model if their influence was significant according to the Wald test (*p* < 0.05) and if they led to a decrease in the unexplained individual variability of the fixed effects.

Model development was guided by the minimum value of the corrected Bayesian information criteria which is penalized by the log of the number of subjects and the log of the total number of observations [16], and the shrinkage in individual parameters estimates (η-shrinkage). The standard errors of the estimated population parameters were calculated via the estimation of the Fisher information matrix. An internal model validation was performed through graphical evaluation of the population and individual observed versus predicted concentrations plots. Simulation-based diagnostics were conducted using prediction-corrected visual predictive check (pcVPC) [17] and normalized prediction distribution errors (NPDEs) [18].

### 2.6. Monte Carlo Simulations

To assess the impact of dose and the different covariates on exposure to roscovitine and its metabolite M3, simulations were performed using the final population PK model. For each scenario, 1000 patients were simulated and the areas under the curve between 0 and 12 h (AUC_0–12h_) were calculated. All simulations were performed by Simulx (mlxR: R package version 4.1; Inria, Paris, France).

## 3. Results

### 3.1. Population Description

Among the 34 subjects included in ROSCO-CF, 23 received roscovitine and were all included in the PK analysis: 9 received 200 mg of roscovitine, 7 received 400 mg, and 7 were given 800 mg. The complete data set included 138 concentrations of roscovitine, of which 19 were below the LLOQ, and 138 concentrations of the M3 metabolite, of which 9 were below the LLOQ (Table 1). The individual concentration profiles are presented as spaghetti plots in Figure 1.

The patient demographics are presented in Table 2.

The ages of the patients ranged from 22 to 51 years, 10 were female and 13 male, and their body mass index ranged from 17 to 32 kg/m^2^. With regard to concomitant PPI treatment, 15 subjects were taking esomeprazole, omeprazole, or pantoprazole during the study. Of the known CYP3A4 inhibitors, 1 patient was taking posaconazole and 14 were taking azithromycin. Genotyping was available for only 20 of the 23 patients tested for CYP2B6, and no DNA could be analyzed for 2 patients. Over 10% of the data were missing for CYP3A4, CYP2B6, and ABCB1 polymorphisms.

### 3.2. Population Pharmacokinetic Model

The structural model that best described the data was a two-compartment parent-metabolite model, with first-order transformation of roscovitine in its carboxylate metabolite M3, without back transformation, and a first-order elimination of M3 (Figure 2).

The elimination of roscovitine could not be identified, its value was extremely small and poorly estimated (%RSE > 100%, relative standard error). In the absence of direct administration of the carboxylate M3, the volumes of distribution of the two molecules could not be distinguished. The volume of the central compartment was modelled by a single apparent parameter common to roscovitine and its metabolite M3, conditioned by the unknown absolute oral bioavailability F of roscovitine. The complex absorption process of roscovitine was described by a sum of two Gaussian inverse density functions for the parent molecule, and one Gaussian inverse density function for the metabolite, evoking a noticeable first-pass effect. Regarding the absorption of roscovitine, in order to avoid instability and overlapping of the different phases of absorption, T2max has been defined as T2max = T1max + dT2max, strictly positive. So as to overcome problems with the identifiability of the carboxylate M3 absorption parameters, the TM3max and CVM3 parameters of the inverse Gaussian distribution were set as equal to those of the first phase of roscovitine absorption—TM3max = T1max and CVM3 = CV1 (Appendix B). The typical value of D50 was estimated to be 1190 µmol, leading to Fr values of 0.32, 0.49, and 0.65, respectively, for the oral doses of 200 mg, 400 mg, and 800 mg roscovitine. The error model that best fit the data was the proportional model for roscovitine, and the combined model for its carboxylate M3. For random effects, substantial inter-individual variability was detected for D50 (68.9%), dT2max (64.7%), and V/F (67.8%). A significant correlation was observed between the inter-individual variability of T1max and CV2 on the one hand, and between those of dT2max and CV1 on the other. With respect to covariates, a substantial influence of PPI use on dT2max was observed, and V/F was significantly associated with the height of the patients. The reference height of the population was 170 cm. The η-shrinkage was less than 30% for all fixed effects for which the inter-individual variability was estimated (Table 3).

The goodness-of-fit plots of the final model are shown in Figure 3 and Figure 4. There was no significant deviation of the NPDE from a normal distribution and the data exhibited no apparent bias in model prediction. According to the pcVPC (Figure 5), the average observed values were well-predicted. Estimates of the population PK parameters are presented in Table 3. The individual fits for roscovitine and its metabolite M3 are provided in Figure 6.

### 3.3. Monte Carlo Simulations

The median (interquartile range) simulated AUC_0–12h_ of roscovitine were 477 ng·h/mL (259–847), 1436 ng·h/mL (850–2327), and 3711 ng·h/mL (2335–6329) for the 200 mg, 400 mg, and 800 mg oral doses, respectively. Regarding the carboxylate M3, the median (interquartile range) simulated AUC_0–12h_ were 1293 ng·h/mL (753–2198), 2557 ng·h/mL (1556–4294), and 4888 ng·h/mL (2893–8559) for the 200 mg, 400 mg and 800 mg oral doses, respectively.

Simulations by dose showed a more-than-proportional increase in plasma exposure to roscovitine, and in parallel a less-than-proportional increase in plasma exposure to the carboxylate M3 with increasing doses of roscovitine (Figure 7). Saturation of the first-pass effect was evidenced by the ratios of AUCs of roscovitine concentrations to AUCs of carboxylate M3 concentrations (AUCrosco/AUCM3) as a function of dose (Figure 8), with an increase in the median AUCrosco/AUCM3 ratio from 0.37 to 0.79 by increasing the dose from 200 mg to 800 mg.

Simulations of the PK profiles of roscovitine and the carboxylate M3 after oral administration of 400 mg roscovitine according to PPI therapy showed the impact of PPIs on the rate of absorption (Figure 9), with no obvious effect on plasma exposure or AUCrosco/AUCM3 ratios (Figure 10).

## 4. Discussion

In this paper we present the first population PK model of roscovitine, and its metabolite M3, which takes into account its complex and highly variable absorption. We have successfully modelled this complex absorption process in CF patients using the sum of inverse Gaussian density with saturable first-pass metabolism. This technique has already been used to model the PK of poorly-soluble drugs with complex absorption [15,19]. This methodology has allowed us to satisfactorily describe the absorption of roscovitine and to study the impact of PPI on the exposure to roscovitine and its carboxylate M3. The inverse Gaussian density function has been previously used as a flexible function to describe the transit time of the absorption phase of different drugs and extravascular routes of administration. The sum of Gaussian inverse densities has been shown to be particularly flexible to describe highly-complex absorption processes, such as those observed after administration of sustained-release forms [20]. However, although this model is useful for studying the absorption process and its variability, it does not provide any indication of the biological mechanisms of absorption. Thus, the model used in this study does not allow us to distinguish between a possible intestinal or hepatic first-pass effect, nor does it allow us to identify the specific contribution of efflux pumps, intestinal and hepatic cytochromes, or the dissolution of roscovitine in biological fluids.

The PK of roscovitine after oral administration is non-linear, as the magnitude of the first-pass effect is dose-dependent. This phenomenon had been observed in pre-clinical studies in rodents and dogs (unpublished pre-clinical acute toxicity studies) and it has been suggested in humans following phase I studies of roscovitine in patients with advanced malignancies [21,22]. In the CF patient population, our Monte Carlo simulations confirmed a more-than-proportional increase in roscovitine AUC for increasing oral doses.

The mechanism of its prolonged absorption is not elucidated. Roscovitine is essentially insoluble in water at pH 7 [23], and slightly soluble at acidic pH, which may contribute to its slow absorption after oral administration. PPI such as omeprazole increase gastric pH by blocking the H⁺/K⁺ APTase of parietal gastric cells. This alkalinization further reduces the solubility of roscovitine and may explain the slower absorption shown in our Monte Carlo simulations. After administration of PPI, Tmax is delayed and Cmax is reduced, as already described for several molecules whose solubility depends on pH [24,25].

To our knowledge, only one compartmental analysis of the PK of roscovitine and its carboxylate M3 has been performed to date. This phase 1 study was conducted in 12 healthy male volunteers who received escalating oral doses of 50 to 800 mg of roscovitine [23]. The model used was comparable to our structural model, with two differences—the tissue distribution of the carboxylate M3 was modelled by a second compartment, not identifiable in our case, and the absorption process was modelled with a lag time. Our analysis corroborates the main results of the study carried out in healthy volunteers. The metabolic elimination of roscovitine was faster than its absorption, and similarly, the carboxylate M3 was characterized by plasma concentrations limited by its rate of formation. The first-pass effect was rapid and large, peak concentrations of roscovitine and its metabolite M3 were concomitant, and the overall PK were characterized by significant inter-individual variability.

Roscovitine is a substrate for CYP3A4, CYP2B6 [26], and the P-glycoprotein 1 (P-gp, MDR1 or ABCB1) efflux pump [27], which probably play an important role in the absorption processes and in particular the first-pass phenomenon, whether hepatic or intestinal. Analysis of the polymorphisms of the genes encoding these proteins does not allow us to explain the importance of inter-individual variability. These results can be partly explained by the small size of our study and the high proportion of missing data in the genomes collected.

Beyond the work presented here, the joint modelling of roscovitine and its metabolite M3 by a population approach will allow the determination of an optimal experimental design for the PK analyses of future clinical trials [28]. It will also open up the possibility of conducting exposure-response studies that will integrate the two molecules, as the contribution of roscovitine and its M3 metabolite to the therapeutic effects studied in humans remains to be elucidated. In addition, several unknowns remain to be studied. First of all, an external validation will be necessary to validate the predictive performance of this model. Secondly, the study of PK after repeated administration will be necessary to evaluate the accumulation of roscovitine and its M3 metabolite in CF patients. Finally, the inter-occasion variability in PK could be greater than the inter-individual variability, as has already been observed for other poorly-soluble molecules [19]; the study of PK after repeated administrations would allow its identification and quantification.

## 5. Conclusions

The PK of roscovitine and its metabolite M3 after oral administration of roscovitine was described by a non-linear, mixed-effect, parent-metabolite model. This modelling revealed a dose-dependent and saturable first-pass effect leading to the presystemic formation of the metabolite M3. In addition, a slower rate of absorption of roscovitine was demonstrated in patients treated with PPIs, although the clinical impact of this reduction in the rate of absorption is uncertain. Overall, the PK of oral roscovitine in adult patients with CF was non-linear and showed significant inter-individual variability.

## Figures and Tables

**Figure 1 pharmaceutics-12-01087-f001:**
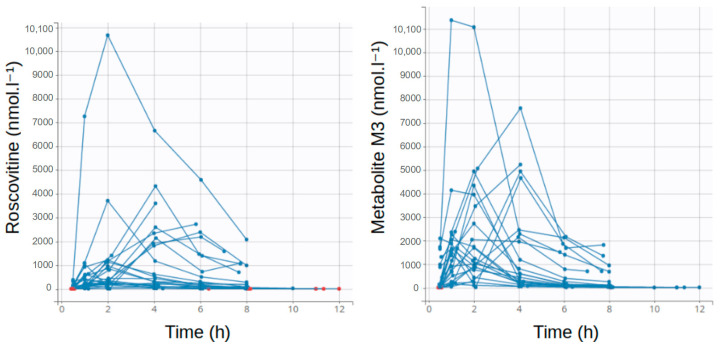
Spaghetti plots of individual profiles of roscovitine and M3 metabolite concentrations.

**Figure 2 pharmaceutics-12-01087-f002:**
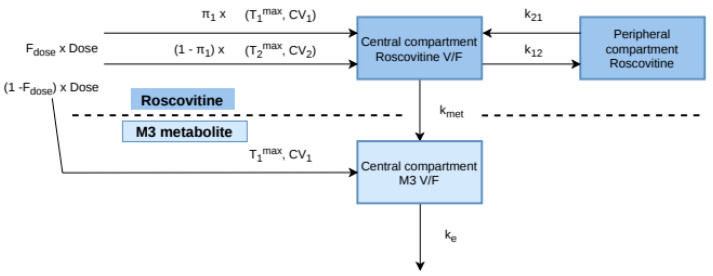
Compartmental model. F_dose_: observed oral bioavailability of roscovitine; π_1_ and (1 − π_1_): weights associated with each input fraction; T_1_^max^, T_2_^max^, CV_1_, and CV_2_: parameters of the inverse Gaussian rates of absorption; V/F: apparent volume of the central compartment after oral administration; k_12_ and k_21_: distribution rates of roscovitine to and from the peripheral compartment; k_met_: systemic conversion rate of roscovitine in its carboxylate metabolite M3; k_e_: elimination rate of the metabolite.

**Figure 3 pharmaceutics-12-01087-f003:**
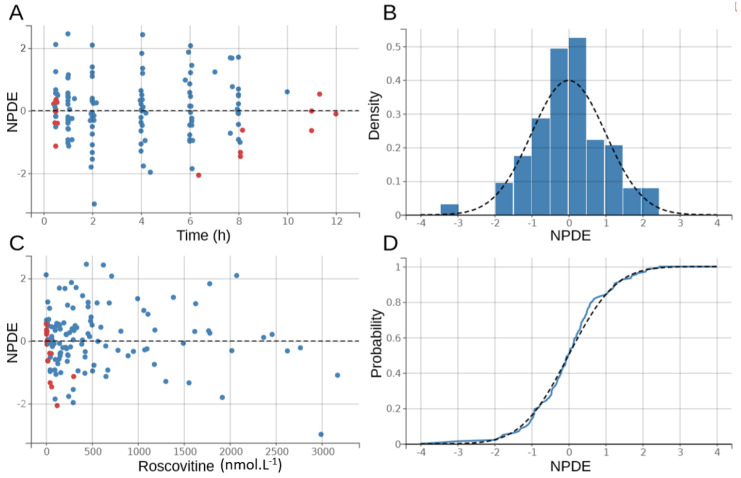
Distribution of the normalized prediction distribution errors (NPDE) for roscovitine. The black dashed line outlines the normal distribution N(0,1); blue dots: NPDE; red dots: NPDE for censored observations. (**A**) NPDE as a function of time in hours, (**B**) density plot of the NPDE for roscovitine, (**C**) NPDE as a function of roscovitine concentrations in nmol·L⁻^1^, and (**D**) QQ-plot of the NPDE for roscovitine.

**Figure 4 pharmaceutics-12-01087-f004:**
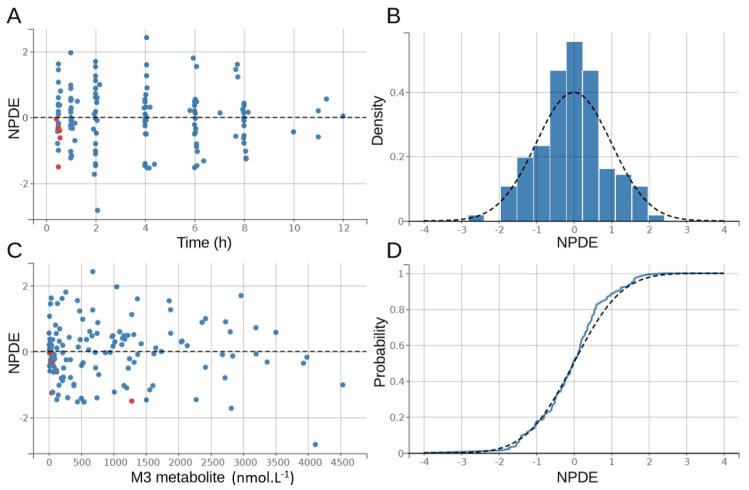
Distribution of the normalized prediction distribution errors (NPDE) for M3. The black dashed line outlines the normal distribution N(0,1); blue dots: NPDE; red dots: NPDE for censored observations. (**A**) NPDE as a function of time in hours, (**B**) density plot of the NPDE for M3, (**C**) NPDE as a function of M3 concentrations in nmol·L⁻^1^, and (**D**) QQ-plot of the NPDE for M3.

**Figure 5 pharmaceutics-12-01087-f005:**
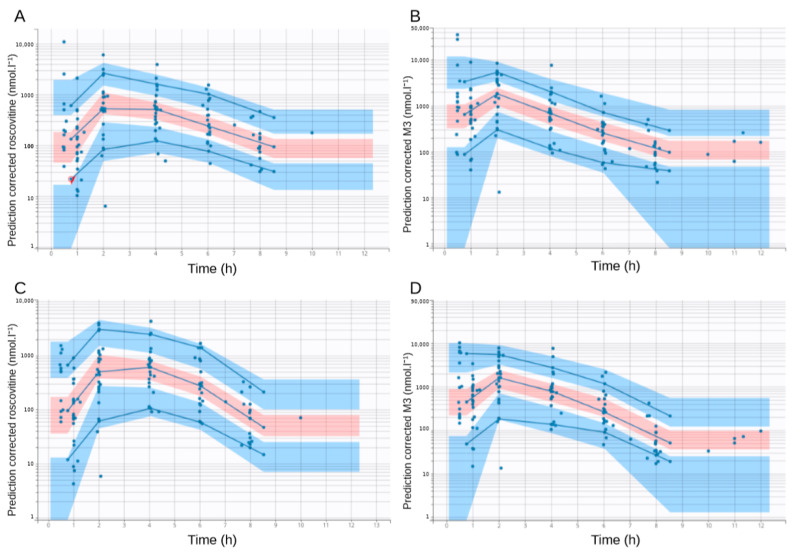
Prediction-corrected visual predictive checks (pcVPC). The red shaded area is the 95% confidence interval (CI) of the median prediction. The blue shaded area represents the 95% CI of the 10th and 90th prediction interval. The blue dots represent the observed data. The blue lines represent the empirical percentiles. (**A**) Prediction-corrected visual predictive check of the final model for roscovitine concentrations over time, (**B**) prediction-corrected visual predictive check of the final model for M3 concentrations over time, (**C**) prediction-corrected visual predictive check of the base population pharmacokinetics model (without covariates) for roscovitine concentrations over time, and (**D**) prediction-corrected visual predictive check of the base population pharmacokinetics model for M3 concentrations over time.

**Figure 6 pharmaceutics-12-01087-f006:**
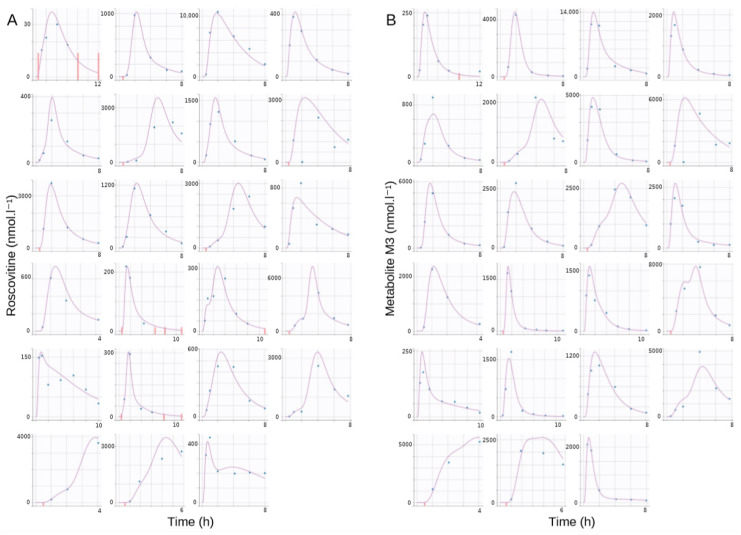
Individual fits: predicted profiles given by the estimated individual model of the final model for (**A**) roscovitine and (**B**) metabolite M3 (empirical Bayes estimates). Blue dots, observed data; purple lines, individual fit; and red bars, censored data.

**Figure 7 pharmaceutics-12-01087-f007:**
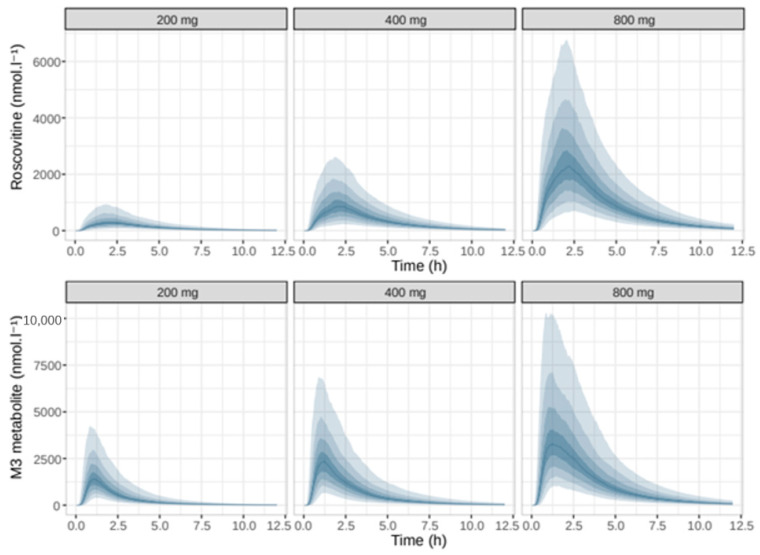
Simulated pharmacokinetic profiles of roscovitine and M3 according to roscovitine oral dose in mg. The solid blue line is the median of the distribution of the simulated concentrations, the shaded areas are the empirical percentiles of the distribution, from 10% to 90%.

**Figure 8 pharmaceutics-12-01087-f008:**
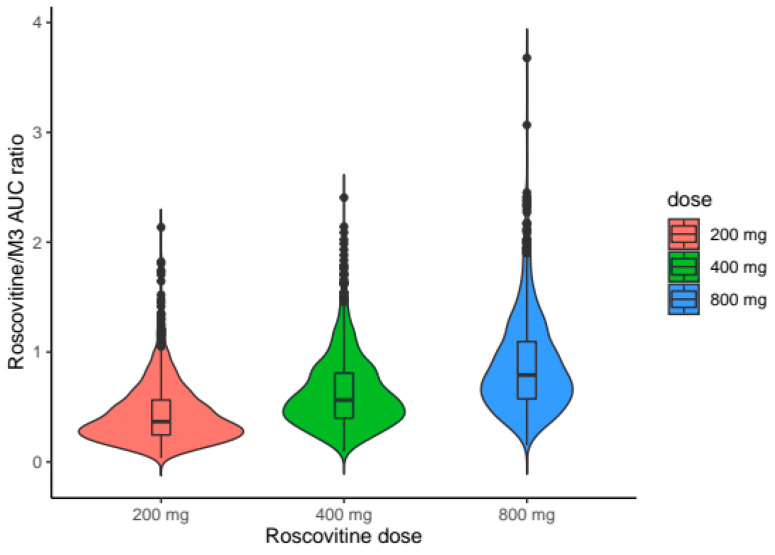
Distribution of the simulated AUC_rosco_/AUC_M3_ ratios according to roscovitine oral dose.

**Figure 9 pharmaceutics-12-01087-f009:**
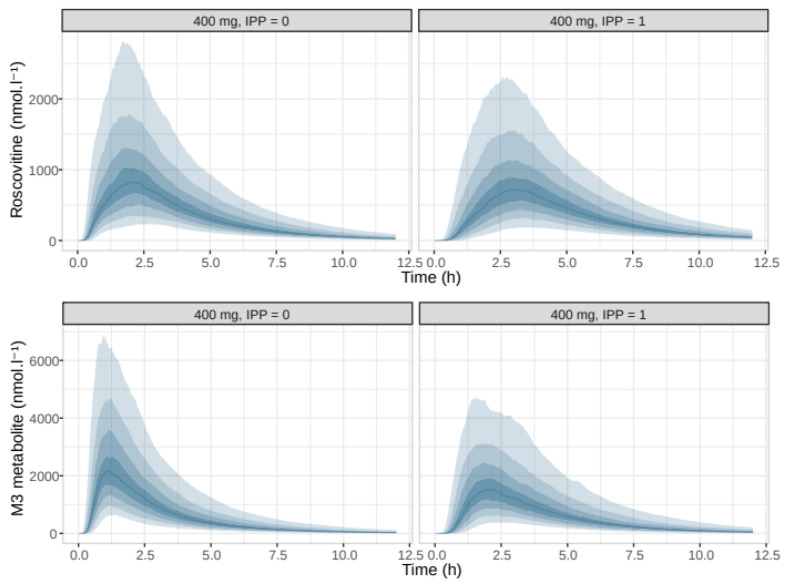
Simulated pharmacokinetic profiles of roscovitine and following a 400 mg oral dose of roscovitine, according to PPI treatment. The solid blue line is the median of the distribution of the simulated concentrations, the shaded areas are the empirical percentiles of the distribution, from 10% to 90%.

**Figure 10 pharmaceutics-12-01087-f010:**
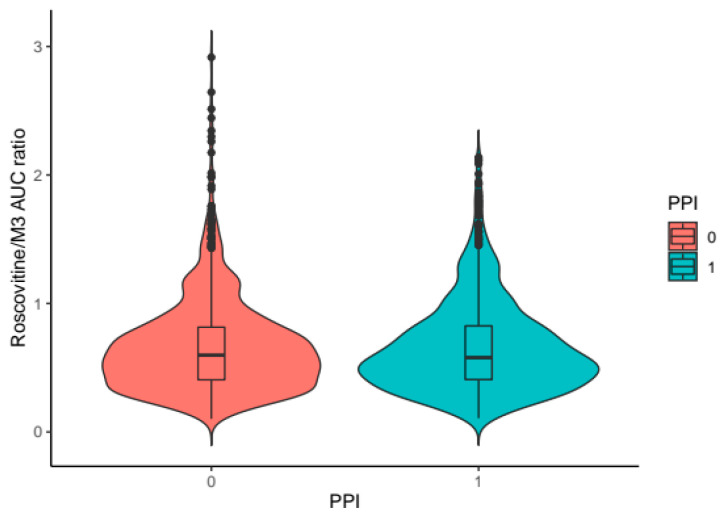
Distribution of the simulated AUC_rosco_/AUC_M3_ ratios according to PPI treatment for a 400 mg roscovitine oral dose.

**Table 1 pharmaceutics-12-01087-t001:** Number of concentration–time data points available for analysis.

Number of Data Points	Global	200 mg Group	400 mg Group	800 mg Group
Total roscovitine	138	57	42	49
• Average per subject	6	6.3	6	5.6
• Minimum per subject	4	4	6	4
• Maximum per subject	7	7	6	6
• Total roscovitine BLQ	19	19	0	0
Total carboxylate metabolite M3	138	57	42	49
• Average per subject	6	6.3	6	5.6
• Minimum per subject	4	4	6	4
• Maximum per subject	7	7	6	6
• Total carboxylate metabolite M3 BLQ	9	9	0	0

BLQ: below limit of quantification, <5.00 ng/mL.

**Table 2 pharmaceutics-12-01087-t002:** Demographic data of the studied population.

	Global	200 mg Group	400 mg Group	800 mg Group
Number of subjects	23	9	7	7
Sex (F)	10	3	4	3
Age (years)	33 (22–51)	32 (22–50)	33 (22–41)	37 (28–51)
Weight (kg)	58 (48–90)	58 (51–90)	56 (49–78)	59 (48–80)
Height (cm)	166 (158–182)	169 (158–182)	165 (158–182)	173 (162–182)
Glomerular filtration rate (mL·min⁻^1^)	113 (61–202)	125 (92–202)	113 (72–144)	107 (61–135)
Proton pump inhibitors	15	4	4	7

Continuous data are presented as median (range).

**Table 3 pharmaceutics-12-01087-t003:** Parameters estimates of the population pharmacokinetics model.

Parameters	Population Parameters Estimates		
	Value	%RSE	%Shrinkage
Fixed Effects			
D_50_ (µmol)	1190	16.5	
π1	0.285	0.378	
T_1_^max^ (h)	0.676	14.6	
β (PPI = 1) on T_1_^max^	0.680	23.4	
dT_2_^max^ (h)	1.04	16.3	
CV_1_	0.542	10.3	
CV_2_	0.354	14.1	
V/F (liters)	62.2	15.4	
β height (cm) on V	6.47	48.3	
k_21_ (h^−1^)	0.768	0.206	
k_12_ (h^−1^)	1.82	0.636	
k_met_ (h^−1^)	2.07	1.13	
k_e_ (h^−1^)	2.58	9.86	
Inter-patient variability standard deviation			
ω D_50_	0.689	18.6	7.83
ω T_1_^max^	0.452	16.2	4.37
ω dT_2_^max^	0.647	23.9	21.1
ω CV_1_	0.426	21.1	24.1
ω CV_2_	0.547	21.7	17.5
ω V/F	0.678	15.7	−2.28
ω k_e_	0.393	19.6	20.5
Correlations between random effects			
Corr. T_1_^max^ CV_2_	−0.839	14	
Corr. dT_2_^max^ CV_1_	0.624	31.8	
Error model parameters			
Roscovitine b1	0.297	10.3	
M3 a2 (nmol·L^−1^)	9.20	22.3	
M3 b2	0.271	10.7	
ε-shrinkage for roscovitine			20.4%
ε-shrinkage for metabolite M3			22.8%

D_50_: oral dose of roscovitine for which F_dose_ is 50%; π_1_: weight associated with the first input fraction; T_1_^max^, dT_2_^max^, CV_1_, and CV_2_: parameters of the inverse Gaussian rates of absorption; β (PPI = 1) on T_1_^max^: coefficient associated with the modification of T_1_^max^ in patients taking PPI; V/F: apparent volume of the central compartment after oral administration; β height (cm) V: coefficient associated with the effect of height on V/F; k_12_ and k_21_: distribution rates of roscovitine to and from the peripheral compartment; k_met_: systemic conversion rate of roscovitine in its carboxylate metabolite M3; k_e_: elimination rate of the metabolite; ω: interpatient variability; roscovitine b1: proportional error for roscovitine concentrations; M3 a2: additive error for M3 concentrations; M3 b2: proportional error for M3 concentrations.

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
