# Peer review of "Non-Linear Pharmacokinetics of Oral Roscovitine (Seliciclib) in Cystic Fibrosis Patients Chronically Infected with Pseudomonas aeruginosa: A Study on Population Pharmacokinetics with Monte Carlo Simulations"

_pharmaceutics, 2020, doi:10.3390/pharmaceutics12111087_

Round 1
Reviewer 1 Report
The authors have presented a population PK model of roscovitine and its metabolite to characterize its complex absorption process. The article adequately describes the methodology and results, but several concerns have been raised, which could improve the overall impact of the article.
-The authors aim to characterize the variable absorption process of roscovitine, but they lack to present a mechanistic or semi-mechanistic model, which would enhance the impact of the final population PK model. Roscovitine has been recognized as a Pgp substrate. The authors are encouraged to explain whether absorption models including Michaelis-Menten kinetics have been considered during model development.
-The use of a sum of inverse Gaussian density functions is purely empirical and does not allow us to distinguish the absorption mechanisms involved in the absorption of roscovitine. The authors should discuss the clinical relevance of this function in the Discussion section to highlight its main limitations.
-An intestinal metabolism is assumed in the final population PK model by incorporating (1-Fdose), which represents the metabolized fraction in the gut. From a kinetic perspective, this model assumes an instantaneous metabolism after dose administration. However, it does not consider the metabolism as a continuous process, which could be affected by the absorption rate of the parent or metabolite. Therefore, the authors should evaluate other strategies to account for the intestinal first-pass effect.
-A relevant aspect of this work is to highlight the fraction of the drug that suffers first-pass metabolism in the gut vs the liver. This information should be mentioned in the Results section.
-Table 3. Eta-shrinkage values should be mentioned with the inter-individual parameters. In addition, eps-shrinkage values should be also incorporated.
-The pc-VPC of the base population PK model should be incorporated to assess the reduction of the inter-individual variability with the final population PK model. Pc-VPC should be represented in semi-log scale.
-A significant bias could be present since LLOQ were discarded. The authors are encouraged to compare the model performance between the current model and the final population PK model incorporating the M3 method for LLOQ.
-A signficant reduction in the rate of absorption is observed in patients receiving PPI drugs. However, it is uncertain whether clinically relevant differences in the exposure metrics would appear and whether an optimal dosing strategy in patients receiving PPI drugs should be performed.
Reviewer 2 Report
The paper completely ignores the physicochemical properties of roscovitine. Nothing about solubility in water, in fasting state simulated intestinal fluid, fed state ..., gastric state. Nothing about the partition coefficient. It is not specified if it was administered roscovitine or a salt of it. Nothing about release kinetics from capsules in different media If roscovitine after dissolution in acidic gastric medium precipitates in intestinal fluid appears a source of variability confounded with and greater than all other sources of variability. It is true that in approximately 95 % of clinical trials these biopharmaceutical factors are ignored, but this approach is not compulsory.
No graphic with all individual plasma levels of roscovotine. No graphic with all individual plasma levels of roscovitine metabolite. A naked eye, a global evaluation would be an additional, very significant information about variability, about possible sharing in sub-clusters, about outliers. In figure 4 appears an outlier subject. Which are the characteristics of the corresponding patient? Which could be the explanation of such a high concentration in plasma?
It is the elimination constant well defined? There are the points on the tail of the curve reliable approximated by a straight line?
With only three points on the ascending part of the curve, it is difficult to determine the parts of three components of absorption mechanisms
With a total of only 23 patients and subgroups very small, can ey speak about population pharmacokinetic?
Reviewer 3 Report
Manuscript -
Non-linear pharmacokinetics of oral Seliciclib (Roscovitine) in cystic fibrosis patients chronically infected with Pseudomonas aeruginosa: a population pharmacokinetics study with Monte Carlo simulations was evaluated for publication in the journal Pharmaceutics and publication of the paper is proposed after a minor revision.
In general, the present study investigated the pharmacokinetics of oral Seliciclib (Roscovitine ) in cystic fibrosis patients chronically infected with Pseudomonas aeruginosa, the results and discussion are adequately structured and written.
Comments: It is suggested that the authors improve/correct these:
Title it is proposed to slightly modify the title: Non-linear pharmacokinetics of oral Seliciclib ( Roscovitine ) in cystic fibrosis patients chronically infected with Pseudomonas aeruginosa: a study on population pharmacokinetics with Monte Carlo simulations.
Line 84: validated liquid chromatography is mentioned, but methodological data are missing, authors are suggested to add a description at LC/MS/ MS. This can also be included in Supplementary Materials or new references can be added.
Line 180: LOD and LOQ data missing.
Lines 335-338, The conclusions are not well written and should be improved.
Round 2
Reviewer 1 Report
The authors have addressed all the issues properly and no other concerns remain. Figure 5 (pc-VPC) has not been included in the revised manuscript.
Reviewer 2 Report
I am enough satisfied by the responses of authors and modifications of the paper